# Perceptions of oral health promotion in primary schools among health and education officials, community leaders, policy makers, teachers, and parents in Gulu district, northern Uganda: A qualitative study

**Peter Akera**[1,2]*, **Sean E. Kennedy**[3], **Aletta E. Schutte**[1,4], **Robyn Richmond**[1],
**Michael Hodgins**[3], **Raghu Lingam**[3‡]

1 School of Population Health, University of New South Wales, Sydney, Australia, 2 School of Clinical Medicine, Faulty of Medicine & Health, University of New South Wales, Sydney, Australia, 3 Faculty of Medicine, Gulu University, Gulu, Uganda, 4 The George Institute for Global Health, Sydney, Australia

☯ These authors contributed equally to this work.
‡ RL is senior author on this work.
* peterakera2010@gmail.com, p.akera@student.unsw.edu.au

**Data Availability Statement:** Data cannot be shared publicly because of restrictions that apply to

## Abstract

### Introduction

One in every two cases of caries in deciduous teeth occurs in low- and middle-income countries (LMICs). The aim of the World Health Organisation's (WHO) Healthy Schools Program is to improve the oral health of children. This study explored perceptions of implementation of the Ugandan oral health schools' program in Gulu district, northern Uganda.

### Methods

Semi-structured interviews were conducted with a purposive sample of 19 participants including health and education officials, community leaders, policy makers, teachers, and parents. All interviews were transcribed verbatim and analysed thematically.

### Results

Our study identified three themes: (1) components of oral health promotion, (2) implementation challenges of oral health promotion, and (3) development of an oral health policy. The components of oral health promotion in schools included engagement of health workers, the community, companies, skills-based education, and oral health services. Participants were concerned about insufficient funding, unsatisfactory skills-based education, and inadequate dental screening. Participants reported that there was an urgent need to develop oral health policy to guide implementation of the program at scale.

the availability of these data, which were used under license for the current study. Data are available from the corresponding author upon reasonable request and with permission of UNSW. The Ethics Committee contact provided may be contacted by fellow researchers for access to the data. The contact information for Human Research Ethics (HREC) Committee at the UNSW is as follows: Phone: +61 2 9385 6222, + 61 2 9385 7257 or + 61 2 9385 7007 Email: humanethics@unsw.edu.au.

**Funding:** The author(s) received no specific funding for this work.

**Competing interests:** The authors have declared that no competing interests exist.

## Conclusions

Schools provided oral health promotion that aligned with existing features of the WHO's health-promoting school framework. Implementation of this strategy could be enhanced with increased resources, adequate oral health education, and explicit development of oral health policy.

## Introduction

The burden of oral diseases remains a substantial population health challenge in low- and middle-income countries (LMICs). In 2017, of the 532 million cases of caries in deciduous teeth observed globally, 265 million were in LMICs [1]. The high prevalence of caries in deciduous teeth has a negative impact on the quality of life of school children. Children with poor oral health suffer from pain and poor oral health negatively impacts activities such as smiling, sleeping, eating, and school attendance [2, 3].

Public health measures such as promoting good oral hygiene, healthy nutritional and behavioural practices, and education about oral diseases [4] reduce the risk of oral diseases. As outlined in the World Health Organisation's (WHO) Healthy Schools Programme, schools provide an optimum location to deliver health promotion activities, where children can develop personal lifelong skills, healthy attitudes, and healthy behaviours, and thereby reduce the risks of oral disease [5].

The Health Promoting School framework incorporates oral health promotion in schools as an integral part of school activities or the curriculum and this supportive environment can also be a channel for interaction with the community. The framework consists of a range of health promotion strategies to improve oral health. These include providing a safe healthy environment to consider oral health, skills-based education, and access to oral health services, improving health promoting policy and practice, and health of the community and engaging health, education, and community leaders [5–7].

There has been limited research in LMICs on the implementation of oral health promotion in schools. Most research has been on oral health status and the associated risk factors for oral diseases among school children [8–11]. In Uganda, one study reported on the impact of establishing four health promoting schools in rural communities in Uganda [12]. That study reported a 26% increase in tooth brushing at least once daily and teachers mentioned the benefits of the program such as a greater awareness about health and fewer absences from school due to 'dental pain'. Key factors identified for success of the program included changes in policy within participating schools, while reported challenges included irregular delivery of supplies. However, that study did not report on factors that increased implementation efficacy nor on factors that would be required to scale up the intervention to a national program.

This study explored the views of key school and community stakeholders on oral health promotion in schools in the Gulu district in Uganda.

## Methods

Semi-structured interviews were conducted with health and education officials, community leaders, policy makers, teachers and parents in Gulu District, Uganda. Gulu district is in the northern part of Uganda about 330 kilometres north of Kampala, the capital city. Over half (53%) of the total population of around 300,000 people are aged between 0 and 17 years.

About a quarter, 26% of persons aged 10–17 years are illiterate. Only 21% of households have access to piped water. The burden of disease in Gulu can be related to poverty, limited access to health facilities and schools, illiteracy, and limited access to clean water.

Interviews with 19 participants were conducted by PA between November 2021 to February 2022. Semi-structured interviews were carried out to gain a deep understanding of the views of health and education officials, community leaders, policy makers, teachers, and parents on the implementation of oral health promotion in schools as they have first-hand knowledge of oral health promotion. The interviews lasted between 15 and 55 minutes. Participants were purposefully chosen to ensure variation regarding rural or urban location of school and occupation, to gain a broad understanding of the diverse contributions to oral health promotion. The district health and education officers provided names of schools and health facilities during physical meetings and phone calls. Head teachers and health facility in-charges provided approval for recruitment of participants from their institution. The names of institutions and description of participants by type of occupation could identify individual participants during or after data collection.

Semi-structured interview guides were developed framed around key aspects of the WHO guidelines on oral health promotion through schools and they were piloted with three participants to gain feedback on the appropriateness of the interview prompts [5–7]. Prompts were used to explore areas of interest to the participants in greater detail. Questions were amended as new themes emerged. Emerging theme patterns and interpretations, and reflections of the research were recorded in memos and analytical notes. All interviews were conducted in-person in offices where there was privacy. All interviews were digitally recorded after informed consent was gained. Participants' responses were transcribed verbatim, accuracy was checked against audio recordings by a person independent of the research team, and transcripts were anonymised.

Data collection and analysis happened concurrently to iteratively refine our interview questions based on emerging codes and themes. Data saturation was reached where no new themes emerged from data analysis; this was discussed with all authors.

## Data analysis

An inductive approach was used for data analysis after transcription of the interviews. Thematic analysis was carried out as described by Terry et al., 2017 [13]. This involved familiarisation and coding of data, theme development, reviewing and defining themes, and developing a report of the themes [13]. Codes were generated by reading the transcripts repeatedly after which we developed candidate themes and refined codes with similar key features. We used a thematic map to identify and understand potential themes. The authors defined and named candidate themes after shaping, clarifying, or even rejecting themes to ensure the themes worked well in relation to the coded data, the dataset, and the research questions. Analysis included writing an analytic narrative that encased the presented data extracts, providing a theme definition; short summaries of the core idea and meaning of each theme. Data and the report were shared with participants to check meaning and interpretation. Key themes and the analytic framework were shared, discussed, and challenged at meetings with all authors. Finally, key themes were compared to Proctor's implementation outcomes [14] as part of a triangulation of our findings.

## Results

Interviews were completed with 19 stakeholders of which six were members of parent-teacher associations (PTA), three members of school management committee (SMC), one city health

officer, one dental surgeon, one district health educator, one education officer, three head teachers and three public health dental officers. Members of PTA, SMC, and head teachers were from five different schools located in northern Uganda.

Three inductive themes were developed from the data. The first theme comprised components of oral health promotion and described how oral health promotion strategies were implemented and comprised three subthemes. The second theme was implementation challenges of oral health promotion and comprised five subthemes describing barriers to successful oral health promotion. The third theme related to the development of oral health policy and comprised four subthemes which analysed participants' perceptions of the development of oral health policy. The themes and subthemes are presented in S1 Table. Presented quotes are illustrative of the theme.

### Theme 1: Components of oral health promotion

The theme, 'components of oral health promotion' comprises the different oral health promotion activities reported by stakeholders and comprised engagement of health workers, the community and companies, skills-based education, and oral health services.

**Subtheme 1: Engagement of health workers, the community, and companies.**   Routine outreach to schools was reported to be conducted by dental health professionals where they provided oral health education, screened children's teeth for decay and treated children for oral diseases and conditions. The outreach was reported to be based on quarterly plans prepared by health workers. "Outreach clinics" were reported to be attended by children, teachers, and community members.

*". . ..basically, from the health facility we arrange (school oral health screening), inform the schools then they mobilise the parents, inform them of what is going to happen at school. We screen and then treat those we can treat."* (Interview 09_Public health dental officer)

Members of the SMC and PTA were engaged in this process. Occasionally the members of SMC and PTA in conjunction with teachers provided oral health education to children in class on brushing of teeth. Parents were also engaged in the program which involved sensitisation of oral hygiene of their children.

*". . ..at school you find that the SMC and the PTA come to school, they get the problem at school level as they go out there, they will be what, ambassadors to tell the community, the parents, that you know, you need to allow your children [to] brush their teeth, rinse teeth and also do flossing."* (Interview 08_Teacher)

Some schools were reported to have yearly programs with toothpaste companies to promote oral health. The activities of the program included distribution of toothpaste and toothbrushes, how-to-brush demonstrations, and oral health awareness. Teachers and students actively participated in the program. This was seen as part of the company's corporate social responsibility.

*". . ..there is also a time we had a program with "an external oral health care organisation". They gave us several toothpaste and toothbrushes so whenever we could go to those schools. . .."* (Interview 02_Public health dental officer)

**Subtheme 2: Skills-based health education.**   The provision of theoretical and practical oral health education was reported to be integrated in regular education by teachers and was

reported to be provided according to the primary school curriculum. Topics included human health, toothbrushing skills, the importance of brushing, and consequences of not brushing such as tooth decay. Teaching methods include lectures, demonstrations, and practical experiences. The curriculum was noted to be delivered by teachers who had received pre- and in-service training on oral health promotion from while at teacher training colleges and non-governmental organisations (NGOs), respectively.

*"We are following the curriculum in health education especially in the issue of the health of the teeth. It is one of the topics that we teach especially in p4 (children aged 10 years). The teaching is done in a classroom and practically the learning aids are given or prepared by the teacher for example we have some samples of toothpaste and toothbrushes."* (Interview 08_Teacher)

**Subtheme 3: Oral health services.**   It was reported by participants that children were screened by teachers during health parades and physical exercise to identify dental caries, poor oral hygiene, signs of oral disease including halitosis, tooth colour change, and other oral health complaints as part of other general health education and care programs. Children were referred by teachers to nearby health facilities for the management of oral health conditions. Some schools required children to have routine medical assessments by different specialists such as opticians, dentists, and ear, nose, and throat surgeons and present a medical report.

*"In some more modern schools when the children are going back for holidays, they give them some medical forms whereby each of them is supposed to go and see a different specialist."* (Interview 02_Public health dental officer)

Health workers from nearby facilities were reported to carry out tooth extractions and atraumatic restorative treatment (ART) during school outreach programs.

## Theme 2: Implementation challenges of oral health promotion

Implementation challenges were assessed as limitations to delivery of oral health promotion in schools. Participants described challenges including insufficient funding, unsatisfactory skills-based education, inadequate dental screening, poor oral health knowledge, and limited parental involvement.

**Subtheme 1: Insufficient funding.**   Participants perceived that oral health was not prioritised by schools and health facilities. This was evidenced by what participants viewed as an insufficient budget for activities and a limited number of NGOs implementing oral health interventions. Limited resources affected the way the schools and health facilities were able to deliver oral health education, the number of school outreached by health workers, and the ability of professionals to treat oral diseases.

*"And since people don't take it [oral health] as a priority, it is always underbudgeted for. . . . . . . .because of limited resources we end up reaching few schools. . . ."* (Interview 14_City health officer)

**Subtheme 2: Unsatisfactory skills-based education.**   Many teachers and health workers felt that oral health education at school was inadequate because the school curriculum only provides general information. There were limited practical sessions, and a lack of knowledge

and skills around oral health. In addition, there is limited in-service training of teachers on oral health education and promotion. There was a lack of educational materials such as charts, posters, and toothbrushes for demonstration of brushing.

*"In the curriculum we have health education, normally these teachers do not concentrate so much on those particular areas they just give general information."* (Interview 05_Headteacher)

**Subtheme 3: Inadequate dental screening.** School staff were considered by participants to lack knowledge in identification, prevention, and treatment of oral diseases. A few schools required children to have periodic assessments from specialist doctors, but dental screening was not among the conditions assessed.

*"In the biodata form there is a provision that shows the allergy thing, maybe the child is suffering from this, should be stipulated by a doctor. About the oral health? Ah no. We only do for the chronic diseases and then we have included the COVID also."* (Interview 07_Headteacher)

For schools that required children to be screened by a dentist, it was reported that the information collected was not used by schools for management of cases. It was suggested that medical assessment of children each term was not working because parents did not want to admit to ill health of their children in case it was somehow detrimental to the child's school progress or reflected badly on them as parents.

*"That one [filling medical forms] is not working because as a parent I have got a good school and I want my child to go to that school, so people always go to a health practitioner they know. So, it is just filled as demanded. But in real sense the child could be having so many dental caries, so people try to hide so that they get access to the school. So that one can only work if it is done from schools but not from home."* (Interview 14_City health officer)

**Subtheme 4: Poor oral health knowledge.** Participants mentioned that the community lacked information around practices to delay development of tooth decay. Information was limited to tooth brushing after meals, but it was reported that parents and children did not know how long a person is supposed to brush their teeth. The population perceived tooth extractions as the only treatment for oral disease and only sought oral health care when they had problems.

*"Most cases if not 90 then 100 percent of dental patients they go to the hospital when they already have problem, none of them will really take their time to go for a check-up."* (Interview 02_Public health dental officer)

Participants also felt that teachers lacked knowledge on providing oral health education.

*"For us who are imparting the knowledge are also unskilled in that field."* (Interview 17_Headteacher)

**Subtheme 5: Limited parental involvement.** Teachers mentioned that parents did not educate their children on oral hygiene or enforce regular cleaning of their children's teeth in the morning before they left for school. Health workers reported that when they conducted

school outreach sessions, parents generally did not attend. Teachers found that when parents were invited to the school to discuss health issues concerning their children, they would often not attend, which in many cases was due to lack of resources or competing demands such as they had to work. Participants mentioned parents spend little time with their children, have limited time for oral health promotion and don't care about their children's oral health as this was not viewed as a priority compared to the day-to-day hardships.

*"Actually we are supposed to meet parents during parents meeting, so generally we talk to parents about a number of issues including the health of their children but unfortunately the type of parents, either because of the economic level that everybody focuses in looking for money, you find they don't even care."* (Interview 05_Headteacher)

## Theme 3: Developing an oral health policy

Participants mentioned the need for the development of policy to guide implementation of oral health promotion in schools. They mentioned policy decisions and actions on factors that affect oral health promotion such as sensitisation of school staff and regular monitoring of the teaching.

**Subtheme 1: Lack of oral health policy.**    Participants mentioned that even though they were trying to implement the Healthy Schools Programme, there was no school-based government oral health policy to guide teachers and health workers. A few noted that the policy that children are required to come to school with toothpaste and toothbrushes was not sufficient if it was not supported by oral health education.

*"We don't have the policy for oral and dental conditions, it is not there."* (Interview-04_District Health Educator)

**Subtheme 2: Policies to improve oral health education.**    Participants felt the need to develop an oral health policy to guide them with implementation of activities to improve the oral health of children. They felt that the oral health policy should encompass oral health education of school staff, regular monitoring of the teaching and learning processes and involvement of parents and oral health professionals.

*". . ..career guidance day could be organised like once a month to have a health talk, they can introduce, they can maybe get a dentist or some other person in the field to go and sensitise (provide health education) them more."* (Interview 02_Public health dental officer)

**Subtheme 3: Improving the paperwork.**    Respondents felt a practical way of improving oral health was to use the oral health school-based clinics for termly screening of children for oral disease. Health workers said the filling of medical forms in private clinics and public health facilities for termly screening was not working because parents would look for health workers they know and have the medical forms filled stating there was no illness present. Health workers said that this method of screening for oral disease did not provide accurate information about the oral health status of the child. A participant mentioned that during the school term children present with advanced stages of disease and yet the medical forms had recently indicated no problems only a short time before.

*"So that is another [policy] area that they should work on that they really need to emphasise that the forms should actually be filled from school not from the clinics when the nurse is there so that we get the true picture, and the child is helped."* (Interview 03_Dental Surgeon)

**Subtheme 4: Parents' responsibilities.**　Participants felt it was essential to increase the responsibility of parents for the oral health of their children though this could be problematic due to the needs of the family and difficulties in the household in areas such as finance. However, an explicit "contract" between the school and the family could increase oral health provision in homes.

> *"Yeah in the oral health policy the key aspect that could be included in the policy which is implementable would be the aspect of brushing, we could emphasize that each parent takes the responsibility of ensuring that the child's oral health is okay by providing the basic necessities that is required."* (Interview 09_Public health dental officer)

## Discussion

This study documented the perceptions of key stakeholders, including those with responsibility for implementing an oral health promotion program in a low-income African setting. Previous research has described various implementation interventions for oral health promotion that resulted in positive oral health outcomes [15–17]. Yet there has been limited work conducted to understand how best to implement oral health promotion programs in schools. Our study identified the components of oral health promotion such as skills-based health education, implementation challenges like insufficient funding, unsatisfactory skills-based education, and inadequate dental screening and development of policy to improve oral health education, paperwork, and parents' responsibilities.

Previous quantitative and mixed methods research identified facilitators for oral health promotion in schools which were capacity and availability of human resources, support and advocacy, and the presence of a policy framework. While the barriers were time constraints, limited involvement, large classes, lack of adequate resources, funding, and collaboration at the local level [18, 19]. While the participants of this study reported many positives in the delivery of this program in Gulu district such as children are taught to use locally available materials to brush their teeth, funding for school outreach, and the presence of persons in charge of health in schools, there were several barriers to implementation that potentially could have hampered program uptake and success. From our thematic analysis we considered our findings in relation to four of Proctor's key implementation outcomes [14]: acceptability, fidelity, appropriateness, and feasibility.

In general, the components of an oral health program this research identified were accepted by stakeholders. Strategies to improve oral health of school children involve interventions that require the engagement of all the stakeholders that we interviewed, including teachers, health and education managers, parents, and community leaders [4, 5]. Participants described good engagement from schoolteachers, parents, and one external organisation. Also, participants commended the preventative nature of the program as a benefit for the community.

In terms of fidelity, we found that participants delivered oral health promotion in alignment with the WHO health promoting school framework with some limitations [5, 6]. Interventions designed according to the WHO's framework have previously resulted in positive oral health outcomes [15–17, 20]. However, insufficient funding and a lack of clear policy guidance for the school proved a challenge to the fidelity of the program. Other studies have similarly noted these implementation challenges [12, 18]. A WHO global survey of school-based health promotion reported inadequate finances, inadequate policy frameworks, and lack of collaboration between local stakeholders as implementation challenges [18]. Other challenges included lack of high-level leadership and governance, poor awareness, attitude, and support among local

users, and failure to provide quality services [18]. In other areas of Uganda, a qualitative study among teachers and key stakeholders of four health promoting schools reported irregular delivery of supplies and a low number of children who participated in the program [12].

Regarding appropriateness, our study demonstrates that delivery of oral health promotion in schools created awareness of oral health for children and the community, children were screened, referred, and treated; children received toothpaste and toothbrushes; and there was active participation by children and teachers. This replicated previous work that reported the benefits from becoming a health-promoting school were greater knowledge and awareness about health, active participation in activities to promote oral health, fewer absences from school due to 'pain' or the need for emergency dental treatment, pride among children and created awareness among community members [12].

The extent to which the oral health program was feasible varied according to participants' reports. While some participants felt that the lack of resources and poor oral health knowledge would hinder oral health promotion, the program engaged the community and companies, provided skills-based education, and oral health services. Our findings indicated the presence of planned outreach initiatives to schools, and to organisations like SMC and PTA for community engagement, collaboration with private companies, a focussed school curriculum and willingness of children to participate in screening, could contribute to successful implementation of such programs. Partnerships, political commitment, timely and accurate communication of information, and evidence-based interventions are necessary for public health programs to succeed [21].

Further research is needed to assess other key implementation issues specifically adoption, sustainability, and penetration of oral health promotion in schools [14]. In addition, as we identified a lack of funding which was repeatedly raised as a barrier to success, an analysis of marginal cost, cost-effectiveness, and cost-benefit would be important for program implementation [14].

The national oral health policy proposes a school oral health program that emphasises implementation of oral health education, screening, and training of teachers in oral health education [22]. However, the national oral health care policy of 2007 is outdated, as it was written over ten years ago. The need for data to inform development of oral health policy has been expressed by researchers and health planners [8, 23].

Future interventions to promote oral health in schools should consider these challenges to implementation. The implication of this study is that health and education officials, community leaders, policy makers, teachers and parents' perceptions of oral health promotion were aligned with the existing local national policy and features of the WHO's health-promoting school framework. However, there is need to provide adequate resources and develop policy for promoting oral health among school children.

## The strengths and limitations of this study

The first author (PA) is a dental surgeon and public health specialist. He has over 20 years' experience in management of oral conditions and oral health promotion in northern Uganda. His knowledge and subject knowledge could have impacted on the questions asked during the interview and how he viewed the data collected. However, as per best practice, he kept a log of his thoughts and feelings during and after the interviews. In addition, themes were discussed with the wider multidisciplinary study team that were non-dental in training.

Health and education officials, community leaders, policy makers, teachers, and parents were experts in their respective fields with several years of experience. This might have influenced how they answered questions towards the perceived "ideal answers" rather than what

they had experienced. However, we purposefully selected a diverse population and report on prominent and unusual themes therefore the study delivers a deep understanding on the views of participants sampled. In addition, purposeful sampling of participants in Gulu District limits generalisability of our findings to other settings in Uganda. Common themes were looked for across all stakeholders and not sub analysis by group, this would need greater numbers in each subgroup. Further limitations are that head teachers may have had bias in providing the list of candidate members PTA and SMC.

## Conclusions

Health and education officials, community leaders, policy makers, teachers and parents play a critical role in program planning, evidence-based decision making and allocation of resources, and ultimately the success of public health interventions. Schools provided oral health promotion that aligned with existing features of the WHO's health-promoting school framework, however a lack of resources, training and practical policy limit the scope and effectiveness of interventions. To improve oral health in the region and other low resource settings, there is need to address implementation challenges such as inadequate resources for oral health promotions and develop oral health policy.

## Supporting information

**S1 Table. Overview of themes and subthemes with illustrative quotes.**
(DOCX)

**S1 Questionnaire. Inclusivity in global research.**
(DOCX)

## Acknowledgments

The authors are grateful to the key stakeholders who participated in the study. We appreciate cooperation of Gulu district and Gulu City administration during the study. We thank the late Professor Mark J Obwolo contributing to the design of the study and for his guidance on data collection.

## Author Contributions

**Conceptualization:** Peter Akera, Sean E. Kennedy, Aletta E. Schutte, Robyn Richmond, Raghu Lingam.

**Formal analysis:** Peter Akera, Sean E. Kennedy, Aletta E. Schutte, Robyn Richmond, Michael Hodgins, Raghu Lingam.

**Investigation:** Peter Akera.

**Methodology:** Peter Akera, Raghu Lingam.

**Project administration:** Peter Akera.

**Resources:** Peter Akera, Raghu Lingam.

**Supervision:** Sean E. Kennedy, Aletta E. Schutte, Robyn Richmond, Raghu Lingam.

**Writing – original draft:** Peter Akera, Sean E. Kennedy, Aletta E. Schutte, Robyn Richmond, Michael Hodgins, Raghu Lingam.

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
