## [Decision Letter · Decision Letter 0]

18 Apr 2023

PONE-D-23-08537Perceptions of oral health promotion in primary schools among health and education officials, community leaders, policy makers, teachers, and parents in Gulu district, northern Uganda: a qualitative studyPLOS ONE

Dear Dr. Akera,

Thank you for submitting your manuscript to PLOS ONE. After careful consideration, we feel that it has merit but does not fully meet PLOS ONE’s publication criteria as it currently stands. Therefore, we invite you to submit a revised version of the manuscript that addresses the points raised during the review process.

We look forward to receiving your revised manuscript.

Kind regards,

Phyllis Lau, PhD

Academic Editor

PLOS ONE

Journal Requirements:

Additional Editor Comments:

This is important research that brings light to an area health promotion that is sometimes neglected or downplayed despite the known link between oral health and systemic health. However, authors need to consider substantial revisions to the manuscript to address the reviewers' comments about qualitative methodology. I also recommend that you use the COREQ checklist to aid your revisions. Finally, we would expect a more thorough proofreading to improve the writing to a publishable standard.

Reviewers' comments:

Reviewer's Responses to Questions

**Comments to the Author**

1. Is the manuscript technically sound, and do the data support the conclusions?

Reviewer #1: Yes

2. Has the statistical analysis been performed appropriately and rigorously? 

Reviewer #1: N/A

3. Have the authors made all data underlying the findings in their manuscript fully available?

Reviewer #1: Yes

4. Is the manuscript presented in an intelligible fashion and written in standard English?

Reviewer #1: No

5. Review Comments to the Author

Reviewer #1: minor grammatical errors have been noted in the manuscript and amendment have been tracked throughout the PDF of the manuscript.

The study is a qualitative study that does not require quantitative statistical analysis. However, some clarification of the qualitative analysis is required. This has been highlighted in the comments provided within the manuscript PDF.

6. PLOS authors have the option to publish the peer review history of their article (what does this mean?). If published, this will include your full peer review and any attached files.

Reviewer #1: **Yes: **Hanny Calache

---

## [Author Response · Author response to Decision Letter 0]

7 Jun 2023

Dear Phyllis Lau

Thank you for your consideration and giving me the opportunity to submit a revised version of our manuscript. 

I have made the changes to the manuscript based on your comments and I am ready to submit the revised version. I have provided a response to revised, revised manuscript with track changes and unmarked version labelled manuscript. 

As a requirement of PLOS One I have included a complete copy of PLOS’ questionnaire on inclusivity in global research and a cover letter addressing legal restrictions on sharing a de-identified data set.

Regards 

Dr. Peter Akera

---

## [Editor Report · Decision Letter 1]

31 Aug 2023

PONE-D-23-08537R1Perceptions of oral health promotion in primary schools among health and education officials, community leaders, policy makers, teachers, and parents in Gulu district, northern Uganda: a qualitative study

PLOS ONE

Dear Dr. Akera,

Thank you for submitting your manuscript to PLOS ONE. After careful consideration, we feel that it has merit but does not fully meet PLOS ONE’s publication criteria as it currently stands. Therefore, we invite you to submit a revised version of the manuscript that addresses the points raised during the review process.

We look forward to receiving your revised manuscript.

Kind regards,

Phyllis Lau, PhD

Academic Editor

PLOS ONE

Journal Requirements:

Additional Editor Comments:

Thank you for responding to the reviewer's extensive comments.

I can understand the reviewers' confusion with regards to the way you have presented some of your themes and subthemes given you made it very clear you conducted inductive analysis, and yet at places you alluded to deductive categorisation. I would like the authors to further address the below:

1. Clearly explain in methods under data analysis how you analyse your data - which part is inductive and when do you apply deductive processes, for instance, using the WHO framework.

2. Please re-address the reviewer comment in the 1st paragraph under Theme 1 - "This is confusing. Were the subthemes of the 'components of health promotion' theme categorised according to the 'WHO Health promoting school framework' or according to the health promotion activities reported by the stakeholders who participated in this study. Please clarify." It is not enough just to clarify in your response, you should edit the text directly to make it clearer.

3. Please address the reviewer comment in the 1st paragraph under Theme 3 - "This is confusing. did you actively seek out information on the development of policies or was this a topic that was raised by the participants. In other words, was 'the assessment of the development of policies to guide the implementation of oral health promotion in schools' an objective of this study or was it a finding from the interview transcriptions?" You have missed this.

4. I appreciate the autors' deletion of Table 1 in response to the reviewers' comments. However, I would prefer table 1 to be attached either as an appendix or as "supplementary materials" and referred to within text. This would help to deminstrate the depth and breath of the data collected.

I look forward to receiving your revised manuscript.

---

## [Author Response · Author response to Decision Letter 1]

12 Oct 2023

Thank you for your comments. We have clearly indicated that the data analysis was inductive, the sub themes of the 'components of health promotion' theme were according to the health promotion activities reported by the stakeholders, the development of policy was a finding from interview transcripts, and included table 1 as supplementary material.

---

## [Editor Report · Decision Letter 2]

19 Oct 2023

Perceptions of oral health promotion in primary schools among health and education officials, community leaders, policy makers, teachers, and parents in Gulu district, northern Uganda: a qualitative study

PONE-D-23-08537R2

Dear Dr. Akera,

We’re pleased to inform you that your manuscript has been judged scientifically suitable for publication and will be formally accepted for publication once it meets all outstanding technical requirements.

Kind regards,

Phyllis Lau, PhD

Academic Editor

PLOS ONE

Additional Editor Comments (optional):

Well done! I congratulate you and your team.
---

## [Editor Report · Acceptance letter]

25 Oct 2023

PONE-D-23-08537R2 

Perceptions of oral health promotion in primary schools among health and education officials, community leaders, policy makers, teachers, and parents in Gulu district, northern Uganda: a qualitative study 

Dear Dr. Akera:

I'm pleased to inform you that your manuscript has been deemed suitable for publication in PLOS ONE. Congratulations! Your manuscript is now with our production department. 

Kind regards, 

on behalf of

Dr. Phyllis Lau 

Academic Editor

PLOS ONE